# Understanding the Vicious Cycle: Relationships between Nonconsensual Sexting Behaviours and Cyberbullying Perpetration

Yunhao Hu , Elizabeth Mary Clancy  and Bianca Klettke *

School of Psychology, Deakin University, Geelong, VIC 3200, Australia
* Correspondence: bianca.klettke@deakin.edu.au

**Abstract:** With the increased ubiquity of digital technology, sexting behaviours, defined as the online sending, receiving, or disseminating of sexually explicit messages, images, or videos, have become increasingly frequent, particularly among young adults. While prior research found sexting behaviours to be associated with cyberbullying behaviours, the role of consent as part of this association has been largely unexplored. The current study investigates whether the relationship between sexting behaviours and cyberbullying perpetration might be explained by a subset of nonconsensual sexting behaviours, such as engagement in nonconsensual sext dissemination and sext-hassling. A large convenience sample of young Western cisgendered adults ($n = 1688$, $M$ age = 23.15, $SD = 3.23$, 52.7% women) completed an anonymous online survey exploring harmful online behaviours (nonconsensual sext dissemination, sext-hassling, cyberbullying victimisation/perpetration). A hierarchical logistic regression was used to analyse predictive relationships between variables. The results showed no significant association between consensual sext-sending and cyberbullying perpetration in young adults. However, nonconsensual sexting behaviours, particularly sext-hassling and nonconsensual sext dissemination, were predictive of cyberbullying perpetration. Finally, cyberbullying victimization appeared to be the most strongly associated factor with cyberbullying perpetration. These findings suggest that future research and prevention efforts surrounding sexting and cyberbullying perpetration would benefit from a focus on consent and the bidirectional nature of cyberbullying behaviours.

**Keywords:** sexting; nonconsensual sexting; cyberbullying; harmful online behaviours; consent; young adult

## 1. Introduction

Advancements in digital technology have created new ways for people to engage in sexual communication, including the online sending, receiving, or disseminating of sexually explicit messages, images, or videos [1], otherwise known as sexting. While consensual sexting is often considered normative in young adults [2,3], it has also been associated with a number of harmful online behaviours such as cyberbullying, defined as intentional and repeated act of aggression carried out through an online medium against people who cannot defend themselves [4]. Prior research has explored the relationship between sexting behaviours and cyber-victimisation [5–7]; however, there is a paucity of information on the sexting predictors associated with cyberbullying perpetration. Further, despite established associations between nonconsensual sexting and psychological distress [8], little is known about the relative impact of nonconsensual sexting on cyberbullying perpetration, since consent has rarely been included in prior studies.

### 1.1. Sexting and Cyberbullying

Sexting has emerged as a common way for people to explore their sexual identities and maintain relations [3]. More than one in ten (10.2% to 14.8%) adolescents [1,9,10], over one-third (38.3%) of emerging adults aged 18–29 [2], and more than half (53.3%) of adults [1],

have engaged in sext-sending behaviours. While sexting is widely regarded as normative behaviour among young people [11,12], it is also associated with risky behaviours such as cyberbullying [13].

Prevalence rates for cyberbullying can be difficult to determine given the lack of consensus regarding a unified definition and/or measurements utilised in current research. A systematic review into cyberbullying behaviour in adults [14] found that rates of victimisation ranged from 2.4% to 90.9%, while perpetration rates ranged from 0.6% to 54.3%. However, given the socially undesirable nature of cyberbullying behaviour, reported rates for perpetration are likely to be underestimates of the actual prevalence. Whilst cyberbullying prevalence is difficult to establish, prior research has found associations with age and gender. A 2016 meta-analytic study [15] encompassing 77 individual studies of adolescents and young adults found that being male and older in age was associated with an increased likelihood of cyberbullying perpetration, whilst other studies presented similar age findings and either similar or no gender differences [16]. However, these findings regarding age, in particular, are somewhat problematic, as the majority of existing studies have focused on adolescents, and adult-specific data is limited, especially beyond young adulthood. One study that specifically investigated cyberbullying in adults found that rates of cyberbullying perpetration decreased with age [14], suggesting that there may be a peak in cyberbullying perpetration in late adolescence/early adulthood.

Findings regarding the potential mental health consequences due to cyberbullying demonstrate more consistency [14,17]. Victims of cyberbullying have been found to experience depression, anxiety, low self-esteem, and suicidal ideation [14,17–19]. Further, cyberbullying perpetration was found to be associated with traditional bullying and intimate partner violence [14,18,20]. Finally, prior literature suggests a reciprocity between cybervictimization and perpetration behaviours [20–23], suggesting that cyberbullying victims may also be involved in perpetration, and vice versa. Given this bidirectional association, it is critical to consider the role of cybervictimization when exploring factors underlying cyberbullying perpetration behaviour.

Empirical research exploring the relationship between sexting and cybervictimization is well established, and several cross-sectional studies have identified a strong association between sexting and cybervictimization [24–27]. These findings are further supported by longitudinal research, which has found that adolescents who sent sexts are more vulnerable to cybervictimization the following year [5,6]. However, despite the wealth of research exploring sexting and cybervictimization, there is little information regarding associations between sexting and cyberbullying perpetration. Available research suggests that sext sending is positively associated with cyberbullying perpetration [13,28], with some studies broadly construing sexting as a form of cyberbullying behaviour [29,30]. Notably, prior studies have failed to differentiate between consensual and nonconsensual forms of sexting.

*1.2. The Role of Consent*

Consent has been found to be critical for distinguishing whether sexting behaviours can be seen as normative and harmless or potentially harmful to mental health [8]. Prior research into sexting and cyberbullying acknowledged the presence of nonconsensual sexting behaviours, such as sext-hassling, defined as the repeated act of requesting someone to send an image-based sext of themselves, or nonconsensual sext dissemination [5], defined as the nonconsensual sharing of received sexts to audiences beyond the intended recipient [31]. However, these nonconsensual behaviours are often considered as risks associated with consensual sexting, rather than examined as a distinct form of harmful behaviour. To the authors' knowledge, the role of consent has not been directly investigated in research regarding sexting and cyberbullying behaviours.

The conflation of consensual and nonconsensual behaviours is problematic for several reasons. Firstly, prior research [8] has established that negative mental health complications are associated specifically with the nonconsensual forms of sexting, rather than sexting more broadly in young adults, and hence these behaviours need to be considered as distinct.

Secondly, if behaviours are conflated in this way, rather than addressing the perpetration of harmful nonconsensual behaviours, the responsibility is placed on potential victims who engage in consensual sexting to avoid risks associated with nonconsensual behaviour (over which they may have little, if any, control) [32,33]. This can result in victim blaming [31,33] and has led to abstinence-based interventions, which have been found to be ineffective in curbing harmful sexting behaviours [11]. To advance the evidence base regarding sexting and cyberbullying behaviours, it is critical to differentiate between consensual and nonconsensual sexting in empirical studies.

### 1.3. Young Adult Populations

An additional gap in the literature is that the vast majority of sexting and cyberbullying research has been based on adolescent samples. This is understandable, given that adolescents are commonly considered to experience higher rates of cybervictimization, and to be more vulnerable to the negative consequences of nonconsensual sexting behaviour than other age groups [24]. However, there are several reasons as to why a focus on young adult cohorts is also warranted.

Firstly, a recent systematic review found that the negative consequences of cyberbullying in adults may be as severe as in younger populations [14]. The review found cyber-victimization to be significantly related to depression in adults, more so than the impacts of in-person bullying and with no differences across gender. Additionally, the long-term effects of cyberbullying include health issues and reduced job satisfaction. Further, a study on victims of cyberbullying found that adolescents experience greater levels of emotional distress when being cyberbullied by adults [17]. Given that communicative technology grants adult perpetrators convenient access to adolescent victims, a more comprehensive understanding of factors that contribute to cyberbullying perpetration in adults is important, as it could help to protect both adolescent and older victims. Finally, given that rates of sexting behaviours for adults are more than five times higher than for adolescents [1], an increased understanding of relationships between sexting and cyberbullying behaviours in adults would have broad applicability for the development of prevention strategies.

### 1.4. Current Research

Prior research on sexting and cyberbullying behaviours focused primarily on cybervictimization, seldomly addressed nonconsensual sexting behaviours, and paid limited consideration to young adult cohorts. The current study adds to existing literature by examining the association between nonconsensual sexting behaviours (sext-hassling, nonconsensual sext dissemination) and cyberbullying perpetration in young adults. Additionally, the study included cybervictimization as a potential explanatory variable for cyberbullying perpetration. After controlling for age and gender, a positive association between sext-sending and cyberbullying perpetration was anticipated (Hypothesis 1). Further, it was hypothesised that nonconsensual forms of sexting, specifically sext-hassling and nonconsensual sext dissemination, would be positively associated with cyberbullying perpetration (Hypothesis 2). Finally, it was anticipated that victims of cyberbullying would be more likely to engage in cyberbullying perpetration (Hypothesis 3).

## 2. Materials and Methods

This study utilised a cross-sectional design to examine associations between age, gender, and a range of online sexting and cyberbehaviours, specifically sext-sending, sext-hassling, nonconsensual sext dissemination, cybervictimization, and cyberbullying perpetration.

### 2.1. Participants

Participants were recruited during two separate survey waves regarding online behaviours conducted between 2020 and 2021. Of the 2828 participants that commenced the survey, 1006 were removed due to failure to complete relevant instruments (e.g., mea-

sures of cyberbullying), 101 were removed for not meeting the inclusion criteria for age, and 33 gender-diverse participants were removed as the sample size was insufficient for analysis, leaving a final analytical sample of 1688 participants.

Participants were young adults aged 18–30 years ($M$ = 23.15, $SD$ = 3.23). With regard to gender, 799 (47.3%) were male, and 889 (52.7%) were female. The majority of participants were from Australia (77.2%), the United Kingdom (UK, 10.0%), or the USA (5.0%). Most reported their ethnicity to be Australian (48.9%), British or European (20.8%), Asian (15.6%), or North American (2.6%); whilst 12.1% of responses reported other ethnicities. Reported sexual orientations included *Heterosexual* (75.7%), *Bisexual* (14.9%), *Lesbian/Gay* (4.3%), and *Uncertain* (2.9%); remaining participants (2.2%) reported *Other*, *Asexual*, or *Prefer not to answer*. More than three quarters (76.4%) of participants were sexually active, and the average active age of becoming sexually active was 17.32 years ($SD$ = 2.41). In terms of educational attainment, most participants had completed a Bachelor's degree (37.7%), followed by Year 12 or equivalent (30.6%), Postgraduate degree (13.4%), Certificate level (6.6%), and Advanced diploma/diploma (6.1%).

*2.2. Procedure*

After obtaining ethics approval from the Deakin University Human Research Ethics Committee, participants were recruited through social media ($n$ = 1206), with additional paid recruitment via the panel aggregator Prolific ($n$ = 482), which was selected based on the high quality of its data [34] Participation was voluntary and confidential, with no incentive offered for general social media participants. Prolific recruitment was targeted towards males to produce a more balanced sample, with participants receiving minor financial reimbursement following survey completion. Advertisements stated that the survey was intended to explore factors that can influence sexting and harmful online behaviours. Those aged 18–30 were encouraged to participate, and anonymity was emphasised. Participants were given a brief online study description and support information in case of any psychological distress. Upon indicating consent, they were directed to the survey, which took approximately 20–25 min to complete.

Demographics for Prolific versus Social Media Participants

On average, the Prolific participants were slightly older (*Prolific Age M* = 24.3, $SD$ = 3.5, *social media Age M* = 22.7, $SD$ = 3.0; $t$ (1686) = 9.34, $p$ < 0.001) and were more likely to be male (*Prolific* 95.4% men, *social media* 28.1% male). Participants recruited through Prolific were more likely to have engaged in sext-hassling (*Prolific* 5.6%, *social media* 3.3%, $\chi^2$ *(1)* = 4.72, $p$ = 0.03), and cyberbullying perpetration behaviours (*Prolific* 39.4%, *social media* 31.3%, $\chi^2$ *(1)* = 10.06, $p$ = 0.002). Participants recruited through social media were more likely to be sexually active (*Prolific* 68.2%, *social media* 79.7%, $\chi^2$ *(1)* = 25.20, $p$ < 0.001), more likely to have sent sexts (*Prolific* 52.3%, *social media* 74.7%, $\chi^2$ *(1)* = 80.01, $p$ < 0.001), and more likely to have disseminated sexts nonconsensually (*Prolific* 3.7%, *social media* 9.4%, $\chi^2$ *(1)* = 15.28, $p$ < 0.001). Prolific and social media participants did not differ significantly in relation to experiencing cybervictimization (*Prolific* 69.3%, *social media* 67.4%, $\chi^2$ *(1)* = 0.56, $p$ = 0.45)

*2.3. Materials*

2.3.1. Demographics

Participants were asked to indicate their gender ("What is your gender identity?", with response options of Man, Woman, or Other), age (in years), sexual orientation (Response options included *Heterosexual/Lesbian/Gay/Bisexual/Uncertain/Asexual/Other/Do not want to say*), country of residence, ethnicity, educational attainment, whether they were sexually active, and their age of first sexual activity (if applicable).

2.3.2. Sexting Behaviours

An online *Sexting Behaviours Questionnaire* based on previous research [Blinded for review] was used to measure sext-sending, sext-hassling, and nonconsensual sext dissem-

ination, in addition to other items, as the data for this study were taken from a larger study. This questionnaire asks participants to report on their experiences of a range of sexting behaviours, including receiving, sending, requesting, disseminating, and receiving disseminated sexts. Relevant questions are described in the following text.

At the start of the questionnaire, sexts were defined as "sexually explicit images, sent, received or shared via mobile phone messaging or apps". Participants were asked whether they had ever sent sexts; "Have you ever sent an image-based sext of yourself?" *(Yes/No)*, or ever requested sexts: "Have you ever asked someone to send you an image-based sext?" *(Yes/No)*. Participants who requested sexts were also asked whether they had ever hassled someone for sexts: "Have you ever hassled someone to send you an image-based sext of themselves via text or mobile app (e.g., asked repeatedly or was pushy)?" *(Yes/No)*. Participants were asked about sext dissemination perpetration: "Have you ever received an image-based sext intended for yourself which you subsequently distributed to another person (this includes showing or sharing the image)?" *(Yes/No)*. Participants who had disseminated were then asked about whether they received consent: "Did you have permission from the person depicted to share this image?" *(Yes/No)*.

### 2.3.3. Cyberbullying Behaviours

In the interests of brevity, the Cyberbullying Victimization/Perpetration Scale [35] was used to measure both cybervictimization and cyberbullying perpetration, as this is a 6-item measure that provides comparable items for both cybervictimization and cyberbullying perpetration. Regarding cybervictimization, participants were asked how many times they had experienced the following in the past year: *received rude or nasty comments from someone while online; become the target of rumours spread online, whether they were true or not;* or *received threatening or aggressive comments while online.* Response options included *everyday/almost everyday; once or twice a week; once or twice a month; a few times a year; less than a few times a year;* or *never.* Participants who reported experiencing any of the three identified scenarios in the previous year were coded as cyberbullying victims.

Analogously, for cyberbullying perpetration, participants were asked how many times they had experienced the following situations in the past year: *made rude or nasty comments to someone while online; spread rumours about someone online, whether they were true or not;* or *made threatening or aggressive comments while online.* The same response options were provided as above. Participants who reported engaging in any of the three behaviours in the previous year were coded as having cyberbullied. In this sample, the Cronbach's alpha was 0.78 for victimisation and 0.75 for perpetration; this was consistent with initial attempts at validation [35], indicating an acceptable level of scale reliability between items.

### *2.4. Analysis*

Descriptive analyses were used to understand key variables, including rates of non-consensual sexting and cyberbullying behaviours. Chi-square analyses of sexting and cyberbullying behaviours were conducted to determine gender patterns. Bivariate correlations were used to explore associations between sexting behaviours, cybervictimisation, and cyberbullying perpetration. Only variables with a significant correlative relationship with cyberbullying perpetration were included in the subsequent stepwise logistic regression.

### 3. Results

Descriptive statistics are provided in Table 1. More than two-thirds of participants (68.3%) engaged in sext-sending, while less than one in ten engaged in nonconsensual sext dissemination (7.8%) and sext-hassling (4.0%). More than two-thirds of participants (68.0%) were victims of cyberbullying abuse, while just over one-third (33.6%) perpetrated cyberbullying behaviour.

**Table 1.** Key Variables by Gender.

| Variable | Full Sample (*n* = 1688) | Males (*n* = 799) | Females (*n* = 889) | Comparison, Male to Female |
|---|---|---|---|---|
| Sext-Sending | 68.3% | 60.3% | 75.5% | $\chi^2$ *(1)* = 44.63, *p* < 0.001 |
| Sext-Hassling | 4.0% | 7.6% | 0.7% | $\chi^2$ *(1)* = 53.47, *p* < 0.001 |
| Nonconsensual Dissemination | 7.8% | 6.5% | 8.9% | $\chi^2$ *(1)* = 3.33, *p* = 0.068 |
| Cyberbully Perpetration | 33.6% | 42.2% | 26.0% | $\chi^2$ *(1)* = 49.42, *p* < 0.001 |
| Cyberbully Victimization | 68.0% | 72.0% | 64.3% | $\chi^2$ *(1)* = 11.23, *p* < 0.001 |

### 3.1. Correlation

Bivariate correlations were used to explore relationships between age (as a continuous variable), gender (as gender was defined as a binary variable, these are point-biserial correlations), sext sending, sext-hassling, nonconsensual dissemination, cybervictimisation, and cyberbullying perpetration (Table 2). Significant positive associations were found between cyberbullying perpetration and the following variables: gender, sext-hassling, nonconsensual dissemination, and cybervictimisation. A significant negative correlation was found between cyberbullying perpetration and age. No significant relationship between cyberbullying perpetration and sext-sending behaviour was found.

**Table 2.** Correlations for Study Variables.

| Variables | 1 | 2 | 3 | 4 | 5 | 6 | 7 |
|---|---|---|---|---|---|---|---|
| 1. Age | 1 | - | | | | | |
| 2. Gender [1] | 0.197 ** | 1 | - | | | | |
| 3. Sext-Sending | −0.020 | −0.163 ** | 1 | - | | | |
| 4. Sext-Hassling | 0.014 | 0.178 ** | 0.106 ** | 1 | - | | |
| 5. Nonconsensual Dissemination | −0.008 | −0.044 | 0.136 ** | 0.077 ** | 1 | - | |
| 6. Cyberbully Perpetration | −0.089 ** | 0.171 ** | 0.024 | 0.138 ** | 0.079 ** | 1 | - |
| 7. Cyberbully Victimisation | −0.117 ** | 0.082 ** | 0.034 | 0.068 ** | 0.047 | 0.419 ** | 1 |

Note. ** *p* < 0.1. [1] Point-biserial correlations for gender, reference category = Females.

### 3.2. Stepwise Logistic Regression

A stepwise logistic regression was conducted to examine the factors were associated with an increased likelihood of cyberbullying perpetration in young adults. Due to the nonsignificant correlation between sext-sending and cyberbullying perpetration, sext-sending was excluded from the regression. A summary of the analysis results is presented in Table 3.

**Table 3.** Logistic Regression Results Regarding Cyberbully Perpetration [1].

| Variable | B | df | *p* | Exp(B) | 95% CI Lower | 95% CI Higher | $R^2_{LL}$ | $R^2_{Change}$ |
|---|---|---|---|---|---|---|---|---|
| Step 1 | | | | | | | 0.036 | 0.036 ** |
| Step 2 | | | | | | | 0.049 | 0.013 ** |
| Step 3 | | | | | | | 0.199 | 0.150 ** |
| Constant | −2.05 | 1 | <0.001 | 0.13 | - | - | | |
| Age | −0.06 | 1 | 0.002 | 0.94 | 0.91 | 0.98 | | |
| Gender [2] | 0.74 | 1 | <0.001 | 2.10 | 1.65 | 2.66 | | |
| Sext-Hassling | 0.98 | 1 | <0.001 | 2.66 | 1.47 | 4.81 | | |
| Nonconsensual Dissemination | 0.57 | 1 | 0.007 | 1.77 | 1.17 | 2.67 | | |
| Cyberbully Victimisation | 2.82 | 1 | <0.001 | 16.69 | 11.02 | 25.28 | | |

Note. ** *p* < 0.01. [1] Results at Step 1 and Step 2 are available on request. [2] Reference category = Females.

Step 1 included age and gender as demographic variables. The model was significant at Step 1 ($\chi^2$ *(2)* = 77.10, *p* < 0.001) and accounted for 3.6% ($R^2_{LL}$ = 0.036) of the variance in cyberbullying perpetration. After controlling for age and gender at Step 1, nonconsensual sext dissemination and sext-hassling behaviours were included in Step 2, which significantly improved the model ($\chi^2$ *(4)* = 105.62, *p* < 0.001), explaining an additional 1.3% ($R^2_{Change}$ = 0.013) of the variance in cyberbullying perpetration.

Step 3 added cybervictimisation as an independent variable. The overall model was significant at Step 3 ($\chi^2$ *(5)* = 429.63, *p* < 0.001), with a correct classification rate of 71.4% and explaining an additional 15.0% ($R^2_{Change}$ = 0.150) of the variance in cyberbullying perpetration. Overall, the model, including demographic, nonconsensual sexting, and cybervictimization variables explained 19.9% ($R^2_{LL}$ = 0.199) of the variance in cyberbullying perpetration. Concerning independent predictors, older participants were slightly less likely to engage in cyberbullying perpetration (*OR* = 0.94, *p* = 0.002), and males were more than twice as likely (*OR* = 2.35, *p* < 0.001) to cyberbully when compared to females. Participants who engaged in sext-hassling were almost three times as likely to engage in cyberbullying perpetration (*OR* = 2.66, *p* < 0.001), while participants who engaged in nonconsensual dissemination were 77% more likely to be cyberbullying perpetrators (*OR* = 1.77, *p* = 0.007). Finally, participants who were cyberbully victims were almost 17 times (*OR* = 16.69, *p* < 0.001) more likely to engage in cyberbullying perpetration. The regression analysis found all independent variables to be significant predictors of cyberbullying perpetration in young adults. The strongest predictor was cybervictimisation, followed by sext-hassling, gender, nonconsensual sext dissemination, and age.

## 4. Discussion

This study aimed to explore the associative relationship between nonconsensual sexting, cybervictimisation, and cyberbullying perpetration. More than two-thirds of participants (68.3%) reported having sent sexts, which is comparable to previous studies [36–38]. However, less than one-tenth of participants reported sext-hassling (4%) and nonconsensual dissemination (7.8%) behaviours. Whilst minimal prior research has explored the prevalence rates for sext-hassling, nonconsensual sext dissemination rates of at least 15% have been reported in previous research [36–38]. This discrepancy may relate to recruitment occurring during COVID-19 isolation (2020–2021) periods, in which many nations imposed restrictions on movement and social gathering to minimise the spread of the Coronavirus epidemic. While some studies found an increase in general sexting prevalence during this time [39–41], limitations to physical intimacy may have decreased rates of nonconsensual sext dissemination, given that over 70% of nonconsensual dissemination occurs through in-person sharing [42].

In relation to cyberbullying, more than two-thirds of participants reported having experienced this, while more than one-third of participants were perpetrators of cyberbullying. Though perpetration rates were substantially lower than victimisation rates, prior studies have found that those who cyberbully tend to do so more than once [20], suggesting that a few perpetrators may be responsible for multiple victims. Although these results fall within the range of previously reported victimisation (2.4% to 90.9%) and perpetration (0.56% to 54.3%) rates for adults [14], select comparison with previous findings is difficult given that research into cyberbullying varies greatly in terms of criteria and measurements, and the published ranges are large. Consistent with previous research [15,18], males were more likely to engage in cyberbullying than women. This study also found that older participants were less likely to engage in cyberbullying perpetration. Whilst prior studies have found that rates of cyberbullying increase with age in adolescent and younger adult samples, this finding is consistent with previous findings [14] showing that cyberbullying in adults declines with age, with lower rates of perpetration occurring outside educational (college) settings.

In contrast to previous research [13,28,43], there was no support for Hypothesis 1, as sext-sending was not found to be associated with cyberbullying perpetration for young

adults. The high rates of sext-sending behaviours found in the present study (68.3%) may suggest the presence of a ceiling effect [44], which may have led to an underestimation of regression parameters, thus making it more difficult to detect the significance. Alternatively, and given the specific findings presented below, this may suggest that sext sending, as a general behaviour, is not specifically associated with cyberbullying perpetration but that nonconsensual sexting behaviours may be more relevant.

As anticipated, nonconsensual sexting behaviours, specifically sext-hassling and nonconsensual sext dissemination, were significantly associated with cyberbullying perpetration, thus supporting Hypothesis 2. This finding lends support to the importance of differentiation and delineation between consensual and nonconsensual sexting behaviours. While these two behaviours seem to be closely aligned in adolescents [29,30], the distinction between consensual and nonconsensual sexting behaviours becomes critical in young adults. During this developmental period, sexting behaviours are more commonly regarded as normative [11,12].

Notably, only nonconsensual sexting behaviours were significantly associated with cyberbullying perpetration. This association could be explained by the conceptual overlap between nonconsensual sexting and cyberbullying perpetration. Prior research identified nonconsensual sexting behaviours, including sext-dissemination, as being associated with intimate partner violence (IPV) [8,45,46]. Similarly, research found cyberbullying perpetration to be associated with IPV in adult populations [14]. It is possible that when considering adult populations, the intersection between nonconsensual sexting and cyberbullying behaviours is also manifested in behaviours associated with IPV. Another possible explanation lies in the normalisation of harmful online behaviours, which can result in a sense that such activities are perhaps less problematic. Prior research into self-endorsed motivations surrounding nonconsensual sext dissemination [38] found that the most common reasons for dissemination were that "it was a joke" or "was not a big deal". Similarly, research into testimonials surrounding perceived motivations for cyberbullying perpetration found that most believed cyberbullying was "no big deal" [36] and done "for fun or for entertainment" [19]. It is possible that people who engage in nonconsensual sexting have a proclivity towards dismissive and harm minimisation attitudes which, in turn, might encourage cyberbullying behaviour. Future research into how such normalization attitudes develop and are reinforced is warranted.

It should be noted that although sext-hassling and nonconsensual sext dissemination were found to be independent predictors and significantly associated with cyberbullying perpetration, they were relatively weak, accounting for less than 10% of the variance. Instead, consistent with findings from previous research [20–23], cybervictimisation was the strongest statistical predictor of cyberbullying perpetration, supporting Hypothesis 3. A potential explanation for this could be that cyberbullying victims retaliate against their perpetrators or redirect negative emotions to other victims, thus fuelling a vicious cycle [21,23]. The strong and bilateral relationship between cyberbullying victims and perpetrators suggests that these behaviours should not be considered in isolation within prevention programs and future research.

### 4.1. Limitations

Several limitations were present within this study. Participant recruitment was conducted through convenience sampling, thus limiting the generalisability. The sample size was relatively large, with a good gender balance; however, online recruitment may have gathered participants who engaged in more frequent online use, thus inflating the prevalence rates of general online behaviours. Further, financial reimbursement for a portion of male participants via Prolific may have impacted the validity of the findings. The occurrence of data collection during periods of COVID-19 isolation also suggests that findings might not be generalisable, given the differences in online behaviour during confinement [39–41].

Additionally, our analysis was cross-sectional, and thus causality cannot be inferred from the findings. Low power in the sample of those who engaged in nonconsensual sexting behaviours, such as sext-hassling ($n = 67$) and nonconsensual dissemination ($n = 131$), presented challenges during data analysis, and replication with a larger sample is recommended as an area of future research. Further, though participant anonymity was emphasised, the socially undesirable nature of harmful behaviours may have led to an underestimate of the prevalence. Finally, this study did not address consensual but unwanted sexting behaviours, such as those performed under coercion or pressure [12]. It is possible that consensual but unwanted sexting may also be a risk factor for cyberbullying behaviour.

*4.2. Future Research*

Despite its limitations, the present study found that nonconsensual sexting behaviour is a risk factor for cyberbullying perpetration in young adults. The study also found that perpetrators of cyberbullying are substantially more likely to also be victimised. The findings suggest that anti-bullying programmes may benefit from discourse surrounding the importance of consent in sexting behaviours for young adults and interventions to build empathy, as many perpetrators may have once been victims of cyberbullying themselves. Future research into nonconsensual sexting and cyberbullying behaviours may benefit from a focus on the bidirectional nature between cyberbullying victimisation and perpetration to better understand whether reciprocity occurs out of revenge, whereby victims retaliate against perpetrators, or an effort to redirect feelings, whereby victims perpetrate new victims [47]. Additionally, in order to better understand how consent is received and whether it is received through coercion or pressure, future studies may investigate nonconsensual sexting behaviours with more nuance through qualitative studies. Finally, in order to provide a more sophisticated understanding of sext-sending and to reduce the likelihood of a ceiling effect, future studies could consider examining the behaviour as a frequency rather than as a binary outcome.

**5. Conclusions**

In contrast to previous research, this study does not indicate a significant link between general sext-sending behaviours and cyberbullying perpetration for young adults. However, a positive association was found between nonconsensual sexting behaviours and cyberbullying perpetration, thus highlighting the critical role of consent in determining what might constitute harmful sexting behaviours for young adults.

Based on our findings, it is recommended that future research should delineate between consensual and nonconsensual sexting behaviours when considering their impacts on harmful behaviours, such as cyberbullying, especially where young adults are concerned. Finally, the strong predictive relationship between cybervictimisation and perpetration reinforces findings from previous research, providing further support for the bi-directional relationship. Prevention efforts towards cyberbullying perpetrators should consider that perpetrators may have previously been victims. Our results suggest that victims often perpetuate the cycle of cyberbullying through retaliation or redirection. As such, approaches to build empathy and understanding for these individuals may be effective in ending the cycle of abuse.

Finally, given these findings, the role of nonconsensual sexting behaviours cannot be underestimated. While these nonconsensual sexting behaviours were shown to have less of an impact than cyberbullying victimisation, they were, nonetheless, independent predictors. As such, intervention and/or prevention efforts of cyberbullying would benefit from a wholistic approach targeting harmful online behaviours such as nonconsensual sexting behaviours, as well as pointing to the reciprocal nature of the behaviour.

**Author Contributions:** Conceptualization, B.K. and Y.H.; methodology, B.K., E.M.C. and Y.H.; software, Y.H.; validation, Y.H., E.M.C. and B.K.; formal analysis, Y.H.; investigation, Y.H., B.K. and E.M.C.; resources, B.K.; data curation, Y.H. and B.K.; writing—original draft preparation.; writing—review and editing, Y.H., E.M.C. and B.K.; visualization, Y.H.; supervision, B.K.; project administration. B.K. and E.M.C.; funding acquisition, B.K. and E.M.C. All authors have read and agreed to the published version of the manuscript.

**Funding:** This research received no external funding. A small departmental grant from Deakin University School of Psychology was provided to support data collection.

**Institutional Review Board Statement:** The study was conducted in accordance with the Declaration of Helsinki, and approved by the Deakin University's Human Research Ethics Committee (Reference 2018-168, approved 20 May 2022).

**Informed Consent Statement:** Informed consent was obtained from all subjects involved in the study.

**Data Availability Statement:** The data that support the findings of this study are available from the corresponding author, [B.K.], upon reasonable request.

**Conflicts of Interest:** The authors declare no conflict of interest.

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
