# Peer review of "Understanding the Vicious Cycle: Relationships between Nonconsensual Sexting Behaviours and Cyberbullying Perpetration"

_sexes, doi:10.3390/sexes4010013_

Round 1

Reviewer 1 Report

Dear authors,

This manuscript addressed an important issue.

Please see my comments across the manuscript.

Reviewer 2 Report

Significant gaps exist in data on the nature, prevalence, impacts and drivers of the various forms of online violence including cyberbullying. Our understanding of and research on online violence is hindered by a lack of clear definitions, tools, and methods, and specifically related to this paper - to measure cyberbullying and consent. Lack of clear measures and tools makes data comparison and interpretation difficult across contexts. As Machado (2022) notes, “Defining these forms of violence represents an ongoing challenge for investigators in understanding the phenomenon in future research.” https://www.mdpi.com/2076-0760/11/5/207

Yet even though these gaps in this area of study are noted by the authors, I don't believe that they have addressed the issue of definition, tools, and measures well.

Specific comments:

1.       The definition of cyberbullying i.e. “…intentional and repeated act of aggression carried out through an online medium against people who cannot defend themselves”, does not consider the victim’s context, motive and outcomes of the abuse, nor does it take into account the link between offline and online abuse – the paper would be strengthened if the authors looked at more current literature on debates and definitions in this field, and broaden the definition to include – psychological, physical, sexual, control and direct aggression (experiences of face to face violence, with a focus on dating violence / IPV).

2.       The key aim of the paper is to understand the link between non-consensual texting and cyberbullying – but there is no discussion on gender inequity, coercive control and other aspects that influence consent. – and there is no discussion on how the study differentiates between consensual and non-consensual sexting / and the challenges of doing so.

3.       In Section 1.3 on adult populations, the authors talk about adolescent samples and young adults as different populations - but I would define young adults as a sub-group of adolescents. Only in the results section do the authors provide an age group for young adults - ie 18-30 years. International agencies (e.g., UNICEF/UN/WHO) statistically define adolescence as aged 10 and 19; and young adulthood to span ages 18 to 26 or 15-24 years – these are all a bit arbitrary, but it is important for studies to explain why and how they selected the age group. In terms of field building, selecting age groups used in other studies helps build comparable data across settings. In the absence of a clear definition up front in the methods makes the paper a confusing read.

4.       On the tool used to collect data on cyberbullying – can the authors discuss more their selection of the scale they used and why more up to date tools were not explored? See: https://www.mdpi.com/2076-0760/11/5/207 for a selection of current cyberbullying tools/questionnaires.

5.       Had the questionnaire been used online before this study? Did you pilot the questionnaire before sending it out? If yes, who with and what adaptations were made as a result? I ask because the incompletion rate of the cyberbullying measures is high.

6.       What is Prolific and why was it chosen? How might this choice have influenced the findings? From a gender equity perspective, it seems odd to pay / incentivise male participation and not afford women the same opportunity. I am sure you debated the ethics of this choice, and it would be good to include some of the discussion and how you made this decision – just a sentence or two.

7.       In terms of the ethics, did the study offer any services post completion of the survey should the questions raise up any issues?

8.       In the analysis, the levels of non-consensual sext sending seem low with women reporting higher levels (NS) – what do other papers find? E.g. https://link.springer.com/article/10.1007/s12147-022-09304-y - the authors’ explanation in the discussion seems a stretch, and maybe a discussion on measures, tools and online survey approach provide an alternative explanation? Ie social confirmation bias (lines 289-292)

The paper needs to engage with more current literature on the topic and build a clear theoretical framework for the analysis that includes the IPV and dating violence literature (gender norms etc.) if it is to help advance our understanding of cyberbullying. 

Reviewer 3 Report

The article presented is well written, presents a theme of public interest and is, without a doubt, of great relevance. For these reasons, I congratulate the authors involved. That said, here are some suggestions for improvement, namely:

- I'm not sure the term "sext-hassling" is widely known so it might be interesting to leave a brief definition when it is first introduced.

- In the section on materials, where socio-demographic measures are mentioned, I believe it is of interest to present the response options for each measure (e.g., for example, on the issue of gender, I am in doubt whether the option "other" or "no" was presented binary", which is relevant, especially considering the non-cis-gender population is vulnerable to phenomena such as bullying);

- In order to facilitate the reading of Table 1, it should be presented all on the same page.

- In the discussion, I am a little apprehensive about the following sentence "However, results are similar to the range of previously reported victimization (2.4% to 90.9%) and perpetration (0.56% to 54.3%) rates for adults", since most of the results would fall within this percentage spectrum. I understand the difficulty of comparison mentioned earlier, but I believe that reformulating this sentence could be beneficial.

Round 2

Reviewer 2 Report

Thank you for addressing my comments.